# Mixup Inference: Better Exploiting Mixup to Defend Adversarial Attacks

**Tianyu Pang**[*]**, Kun Xu**[*]**, Jun Zhu**[†]
Dept. of Comp. Sci. & Tech., BNRist Center, Institute for AI, Tsinghua University; RealAI
{pty17,xu-k16}@mails.tsinghua.edu.cn, dcszj@tsinghua.edu.cn

## Abstract

It has been widely recognized that adversarial examples can be easily crafted to fool deep networks, which mainly root from the locally unreasonable behavior nearby input examples. Applying mixup in training provides an effective mechanism to improve generalization performance and model robustness against adversarial perturbations, which introduces the *globally linear behavior* in-between training examples. However, in previous work, the mixup-trained models only passively defend adversarial attacks in inference by directly classifying the inputs, where the induced global linearity is not well exploited. Namely, since the locality of the adversarial perturbations, it would be more efficient to actively break the locality via the globality of the model predictions. Inspired by simple geometric intuition, we develop an inference principle, named **mixup inference (MI)**, for mixup-trained models. MI mixups the input with other random clean samples, which can shrink and transfer the equivalent perturbation if the input is adversarial. Our experiments on CIFAR-10 and CIFAR-100 demonstrate that MI can further improve the adversarial robustness for the models trained by mixup and its variants.

## 1 Introduction

Deep neural networks (DNNs) have achieved state-of-the-art performance on various tasks (Goodfellow et al., 2016). However, counter-intuitive adversarial examples generally exist in different domains, including computer vision (Szegedy et al., 2014), natural language processing (Jin et al., 2019), reinforcement learning (Huang et al., 2017), speech (Carlini & Wagner, 2018) and graph data (Dai et al., 2018). As DNNs are being widely deployed, it is imperative to improve model robustness and defend adversarial attacks, especially in safety-critical cases. Previous work shows that adversarial examples mainly root from the locally unstable behavior of classifiers on the data manifolds (Goodfellow et al., 2015; Fawzi et al., 2016; 2018; Pang et al., 2018b), where a small adversarial perturbation in the input space can lead to an unreasonable shift in the feature space.

On the one hand, many previous methods try to solve this problem in the inference phase, by introducing transformations on the input images. These attempts include performing local linear transformation like adding Gaussian noise (Tabacof & Valle, 2016), where the processed inputs are kept nearby the original ones, such that the classifiers can maintain high performance on the clean inputs. However, as shown in Fig. 1(a), the equivalent perturbation, i.e., the crafted adversarial perturbation, is still $\delta$ and this strategy is easy to be adaptively evaded since the randomness of $\overline{x}_0$ w.r.t $x_0$ is local (Athalye et al., 2018). Another category of these attempts is to apply various non-linear transformations, e.g., different operations of image processing (Guo et al., 2018; Xie et al., 2018; Raff et al., 2019). They are usually off-the-shelf for different classifiers, and generally aim to disturb the adversarial perturbations, as shown in Fig. 1(b). Yet these methods are not quite reliable since there is no illustration or guarantee on to what extent they can work.

On the other hand, many efforts have been devoted to improving adversarial robustness in the training phase. For examples, the adversarial training (AT) methods (Madry et al., 2018; Zhang et al., 2019; Shafahi et al., 2019) induce locally stable behavior via data augmentation on adversarial examples. However, AT methods are usually computationally expensive, and will often degenerate model

---

[*]Equal contribution. [†]Corresponding author.

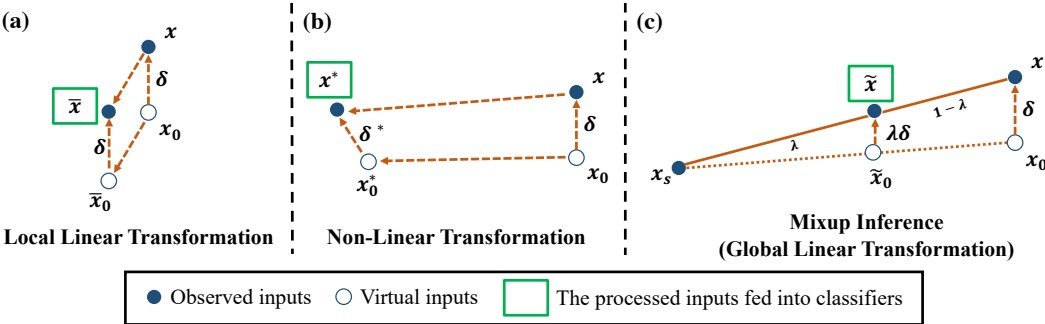

Figure 1: Intuitive mechanisms in the input space of different input-processing based defenses. $x$ is the crafted adversarial example, $x_0$ is the original clean example, which is virtual and unknown for the classifiers. $\delta$ is the adversarial perturbation.

performance on the clean inputs or under general-purpose transformations like rotation (Engstrom et al., 2019). In contrast, the mixup training method (Zhang et al., 2018) introduces *globally linear behavior* in-between the data manifolds, which can also improve adversarial robustness (Zhang et al., 2018; Verma et al., 2019a). Although this improvement is usually less significant than it resulted by AT methods, mixup-trained models can keep state-of-the-art performance on the clean inputs; meanwhile, the mixup training is computationally more efficient than AT. The interpolated AT method (Lamb et al., 2019) also shows that the mixup mechanism can further benefit the AT methods.

However, most of the previous work only focuses on embedding the mixup mechanism in the training phase, while the induced global linearity of the model predictions is not well exploited in the inference phase. Compared to passive defense by directly classifying the inputs (Zhang et al., 2018; Lamb et al., 2019), it would be more effective to actively defend adversarial attacks by breaking their locality via the globally linear behavior of the mixup-trained models. In this paper, we develop an inference principle for mixup-trained models, named **mixup inference (MI)**. In each execution, MI performs a global linear transformation on the inputs, which mixups the input $x$ with a sampled clean example $x_s$, i.e., $\tilde{x} = \lambda x + (1 - \lambda)x_s$ (detailed in Alg. 1), and feed $\tilde{x}$ into the classifier as the processed input.

There are two basic mechanisms for robustness improving under the MI operation (detailed in Sec. 3.2.1), which can be illustrated by simple geometric intuition in Fig. 1(c). One is *perturbation shrinkage*: if the input is adversarial, i.e., $x = x_0 + \delta$, the perturbation $\delta$ will shrink by a factor $\lambda$ after performing MI, which is exactly the mixup ratio of MI according to the similarity between triangles. Another one is *input transfer*: after the MI operation, the reduced perturbation $\lambda\delta$ acts on random $\tilde{x}_0$. Comparing to the spatially or semantically local randomness introduced by Gaussian noise or image processing, $\tilde{x}_0$ introduces spatially global and semantically diverse randomness w.r.t $x_0$. This makes it less effective to perform adaptive attacks against MI (Athalye et al., 2018). Furthermore, the global linearity of the mixup-trained models ensures that the information of $x_0$ remained in $\tilde{x}_0$ is proportional to $\lambda$, such that the identity of $x_0$ can be recovered from the statistics of $\tilde{x}_0$.

In experiments, we evaluate MI on CIFAR-10 and CIFAR-100 (Krizhevsky & Hinton, 2009) under the *oblivious attacks* (Carlini & Wagner, 2017) and the *adaptive attacks* (Athalye et al., 2018). The results demonstrate that our MI method is efficient in defending adversarial attacks in inference, and is also compatible with other variants of mixup, e.g., the interpolated AT method (Lamb et al., 2019). Note that Shimada et al. (2019) also propose to mixup the input points in the test phase, but they do not consider their method from the aspect of adversarial robustness.

## 2 PRELIMINARIES

In this section, we first introduce the notations applied in this paper, then we provide the formula of mixup in training. We introduce the adversarial attacks and threat models in Appendix A.1.

### 2.1 NOTATIONS

Given an input-label pair $(x, y)$, a classifier $F$ returns the softmax prediction vector $F(x)$ and the predicted label $\hat{y} = \arg\max_{j \in [L]} F_j(x)$, where $L$ is the number of classes and $[L] = \{1, \cdots, L\}$. The classifier $F$ makes a correct prediction on $x$ if $y = \hat{y}$. In the adversarial setting, we augment the

data pair $(x, y)$ to a triplet $(x, y, z)$ with an extra binary variable $z$, i.e.,

$$z = \begin{cases} 1, & \text{if } x \text{ is adversarial,} \\ 0, & \text{if } x \text{ is clean.} \end{cases} \tag{1}$$

The variable $z$ is usually considered as hidden in the inference phase, so an input $x$ (either clean or adversarially corrupted) can be generally denoted as $x = x_0 + \delta \cdot \mathbf{1}_{z=1}$. Here $x_0$ is a clean sample from the data manifold $p(x)$ with label $y_0$, $\mathbf{1}_{z=1}$ is the indicator function, and $\delta$ is a potential perturbation crafted by adversaries. It is worthy to note that the perturbation $\delta$ should not change the true label of the input, i.e., $y = y_0$. For $\ell_p$-norm adversarial attacks (Kurakin et al., 2017; Madry et al., 2018), we have $\|\delta\|_p \leq \epsilon$, where $\epsilon$ is a preset threshold. Based on the assumption that adversarial examples are off the data manifolds, we formally have $x_0 + \delta \notin \text{supp}(p(x))$ (Pang et al., 2018a).

## 2.2 MIXUP IN TRAINING

In supervised learning, the most commonly used training mechanism is the empirical risk minimization (ERM) principle (Vapnik, 2013), which minimizes $\frac{1}{n} \sum_{i=1}^n \mathcal{L}(F(x_i), y_i)$ on the training dataset $\mathcal{D} = \{(x_i, y_i)\}_{i=1}^n$ with the loss function $\mathcal{L}$. While computationally efficient, ERM could lead to memorization of data (Zhang et al., 2017) and weak adversarial robustness (Szegedy et al., 2014).

As an alternative, Zhang et al. (2018) introduce the mixup training mechanism, which minimizes $\frac{1}{m} \sum_{j=1}^m \mathcal{L}(F(\tilde{x}_j), \tilde{y}_j)$. Here $\tilde{x}_j = \lambda x_{j0} + (1-\lambda)x_{j1}$; $\tilde{y}_j = \lambda y_{j0} + (1-\lambda)y_{j1}$, the input-label pairs $(x_{j0}, y_{j0})$ and $(x_{j1}, y_{j1})$ are randomly sampled from the training dataset, $\lambda \sim \text{Beta}(\alpha, \alpha)$ and $\alpha$ is a hyperparameter. Training by mixup will induce globally linear behavior of models in-between data manifolds, which can empirically improve generalization performance and adversarial robustness (Zhang et al., 2018; Tokozume et al., 2018a;b; Verma et al., 2019a;b). Compared to the adversarial training (AT) methods (Goodfellow et al., 2015; Madry et al., 2018), trained by mixup requires much less computation and can keep state-of-the-art performance on the clean inputs.

## 3 METHODOLOGY

Although the mixup mechanism has been widely shown to be effective in different domains (Berthelot et al., 2019; Beckham et al., 2019; Verma et al., 2019a;b), most of the previous work only focuses on embedding the mixup mechanism in the training phase, while in the inference phase the global linearity of the trained model is not well exploited. Compared to passively defending adversarial examples by directly classifying them, it would be more effective to actively utilize the globality of mixup-trained models in the inference phase to break the locality of adversarial perturbations.

### 3.1 MIXUP INFERENCE

The above insight inspires us to propose the **mixup inference (MI)** method, which is a specialized inference principle for the mixup-trained models. In the following, we apply colored $y$, $\hat{y}$ and $y_s$ to visually distinguish different notations. Consider an input triplet $(x, y, z)$, where $z$ is unknown in advance. When directly feeding $x$ into the classifier $F$, we can obtain the predicted label $\hat{y}$. In the adversarial setting, we are only interested in the cases where $x$ is correctly classified by $F$ if it is clean, or wrongly classified if it is adversarial (Kurakin et al., 2018). This can be formally denoted as

$$\mathbf{1}_{y \neq \hat{y}} = \mathbf{1}_{z=1}. \tag{2}$$

The general mechanism of MI works as follows. Every time we execute MI, we first sample a label $y_s \sim p_s(y)$, then we sample $x_s$ from $p_s(x|y_s)$ and mixup it with $x$ as $\tilde{x} = \lambda x + (1-\lambda)x_s$. $p_s(x, y)$ denotes the sample distribution, which is constrained to be on the data manifold, i.e., $\text{supp}(p_s(x)) \subset \text{supp}(p(x))$. In practice, we execute MI for $N$ times and average the output predictions to obtain $F_{\text{MI}}(x)$, as described in Alg. 1. Here we fix the mixup ratio $\lambda$ in MI as a hyperparameter, while similar properties hold if $\lambda$ comes from certain distribution.

### 3.2 THEORETICAL ANALYSES

Theoretically, with unlimited capability and sufficient clean samples, a well mixup-trained model $F$ can be denoted as a linear function $H$ on the convex combinations of clean examples (Hornik et al., 1989; Guo et al., 2019), i.e., $\forall x_i, x_j \sim p(x)$ and $\lambda \in [0, 1]$, there is

$$H(\lambda x_i + (1-\lambda)x_j) = \lambda H(x_i) + (1-\lambda)H(x_j). \tag{3}$$

---

**Algorithm 1** Mixup Inference (MI)

---

**Input:** The mixup-trained classifier $F$; the input $x$.
**Hyperparameters:** The sample distribution $p_s$; the mixup ratio $\lambda$; the number of execution $N$.
Initialize $F_{\text{MI}}(x) = \mathbf{0}$;
**for** $k = 1$ **to** $N$ **do**
    Sample $y_{s,k} \sim p_s(y_s)$, $x_{s,k} \sim p_s(x_s | y_{s,k})$;
    Mixup $x$ with $x_{s,k}$ as $\tilde{x}_k = \lambda x + (1 - \lambda) x_{s,k}$;
    Update $F_{\text{MI}}(x) = F_{\text{MI}}(x) + \frac{1}{N} F(\tilde{x}_k)$;
**end for**
**Return:** The prediction $F_{\text{MI}}(x)$ of input x.

---

Specially, we consider the case where the training objective $\mathcal{L}$ is the cross-entropy loss, then $H(x_i)$ should predict the one-hot vector of label $y_i$, i.e., $H_y(x_i) = \mathbf{1}_{y=y_i}$. If the input $x = x_0 + \delta$ is adversarial, then there should be an extra non-linear part $G(\delta; x_0)$ of $F$, since $x$ is off the data manifolds. Thus for any input $x$, the prediction vector can be compactly denoted as

$$F(x) = F(x_0 + \delta \cdot \mathbf{1}_{z=1}) = H(x_0) + G(\delta; x_0) \cdot \mathbf{1}_{z=1}. \tag{4}$$

According to Eq. (3) and Eq. (4), the output of $\tilde{x}$ in MI is given by:

$$\begin{aligned} F(\tilde{x}) &= H(\tilde{x}_0) + G(\lambda \delta; \tilde{x}_0) \cdot \mathbf{1}_{z=1} \\ &= \lambda H(x_0) + (1 - \lambda) H(x_s) + G(\lambda \delta; \tilde{x}_0) \cdot \mathbf{1}_{z=1}, \end{aligned} \tag{5}$$

where $\tilde{x}_0 = \lambda x_0 + (1 - \lambda) x_s$ is a virtual unperturbed counterpart of $\tilde{x}$ as shown in Fig. 1(c). Note that $F_{\text{MI}}(x)$ in Alg. 1 is a Monte Carlo approximation of $\mathbb{E}_{p_s}[F(\tilde{x})]$ as

$$F_{\text{MI}}(x) = \frac{1}{N} \sum_{i=1}^{N} F(\tilde{x}_i) \xrightarrow{\infty} \mathbb{E}_{p_s}[F(\tilde{x})], \tag{6}$$

where $\xrightarrow{\infty}$ represents the limitation when the execution times $N \to \infty$. Now we separately investigate the $y$-th and $\hat{y}$-th (could be the same one) components of $F(\tilde{x})$ according to Eq. (5), and see how these two components differ from those of $F(x)$. These two components are critical because they decide whether we can correctly classify or detect adversarial examples (Goodfellow et al., 2016). Note that there is $H_y(x_0) = 1$ and $H_{y_s}(x_s) = 1$, thus we have the $y$-th components as

$$\begin{aligned} F_y(x) &= 1 + G_y(\delta; x_0) \cdot \mathbf{1}_{z=1}; \\ F_y(\tilde{x}) &= \lambda + (1 - \lambda) \cdot \mathbf{1}_{y=y_s} + G_y(\lambda \delta; \tilde{x}_0) \cdot \mathbf{1}_{z=1}. \end{aligned} \tag{7}$$

Furthermore, according to Eq. (2), there is $\mathbf{1}_{y=\hat{y}} = \mathbf{1}_{z=0}$. We can represent the $\hat{y}$-th components as

$$\begin{aligned} F_{\hat{y}}(x) &= \mathbf{1}_{z=0} + G_{\hat{y}}(\delta; x_0) \cdot \mathbf{1}_{z=1}; \\ F_{\hat{y}}(\tilde{x}) &= \lambda \cdot \mathbf{1}_{z=0} + (1 - \lambda) \cdot \mathbf{1}_{\hat{y}=y_s} + G_{\hat{y}}(\lambda \delta; \tilde{x}_0) \cdot \mathbf{1}_{z=1}. \end{aligned} \tag{8}$$

From the above formulas we can find that, except for the hidden variable $z$, the sampling label $y_s$ is another variable which controls the MI output $F(\tilde{x})$ for each execution. Different distributions of sampling $y_s$ result in different versions of MI. Here we consider two easy-to-implement cases:

**MI with predicted label (MI-PL):** In this case, the sampling label $y_s$ is the same as the predicted label $\hat{y}$, i.e., $p_s(y) = \mathbf{1}_{y=\hat{y}}$ is a Dirac distribution on $\hat{y}$.

**MI with other labels (MI-OL):** In this case, the label $y_s$ is uniformly sampled from the labels other than $\hat{y}$, i.e., $p_s(y) = \mathcal{U}_{\hat{y}}(y)$ is a discrete uniform distribution on the set $\{y \in [L] | y \neq \hat{y}\}$.

We list the simplified formulas of Eq. (7) and Eq. (8) under different cases in Table 1 for clear representation. With the above formulas, we can evaluate how the model performance changes with and without MI by focusing on the formula of

$$\Delta F(x; p_s) = F_{\text{MI}}(x) - F(x) \xrightarrow{\infty} \mathbb{E}_{p_s}[F(\tilde{x})] - F(x). \tag{9}$$

Specifically, in the general-purpose setting where we aim to correctly classify adversarial examples (Madry et al., 2018), we claim that the MI method improves the robustness if the prediction

Table 1: The the simplified formulas of Eq. (7) and Eq. (8) in different versions of MI. Here MI-PL indicates mixup inference with predicted label; MI-OL indicates mixup inference with other labels.

| | **MI-PL** | | **MI-OL** | |
|---|---|---|---|---|
| | $z=0$ | $z=1$ | $z=0$ | $z=1$ |
| $F_y(x)$ | 1 | $1 + G_y(\delta; x_0)$ | 1 | $1 + G_y(\delta; x_0)$ |
| $F_y(\tilde{x})$ | 1 | $\lambda + G_y(\lambda\delta; \tilde{x}_0)$ | $\lambda$ | $\lambda + (1-\lambda) \cdot \mathbf{1}_{y=y_s} + G_y(\lambda\delta; \tilde{x}_0)$ |
| $F_{\hat{y}}(x)$ | 1 | $G_{\hat{y}}(\delta; x_0)$ | 1 | $G_{\hat{y}}(\delta; x_0)$ |
| $F_{\hat{y}}(\tilde{x})$ | 1 | $(1-\lambda) + G_{\hat{y}}(\lambda\delta; \tilde{x}_0)$ | $\lambda$ | $G_{\hat{y}}(\lambda\delta; \tilde{x}_0)$ |

value on the true label $y$ increases while it on the adversarial label $\hat{y}$ decreases after performing MI when the input is adversarial ($z = 1$). This can be formally denoted as

$$\Delta F_y(x; p_s)|_{z=1} > 0; \quad \Delta F_{\hat{y}}(x; p_s)|_{z=1} < 0. \tag{10}$$

We refer to this condition in Eq. (10) as **robustness improving condition (RIC)**. Further, in the detection-purpose setting where we want to detect the hidden variable $z$ and filter out adversarial inputs, we can take the gap of the $\hat{y}$-th component of predictions before and after the MI operation, i.e., $\Delta F_{\hat{y}}(x; p_s)$ as the detection metric (Pang et al., 2018a). To formally measure the detection ability on $z$, we use the **detection gap (DG)**, denoted as

$$\mathbb{DG} = \Delta F_{\hat{y}}(x; p_s)|_{z=1} - \Delta F_{\hat{y}}(x; p_s)|_{z=0}. \tag{11}$$

A higher value of DG indicates that $\Delta F_{\hat{y}}(x; p_s)$ is better as a detection metric. In the following sections, we specifically analyze the properties of different versions of MI according to Table 1, and we will see that the MI methods can be used and benefit in different defense strategies.

### 3.2.1 MIXUP INFERENCE WITH PREDICTED LABEL

In the MI-PL case, when the input is clean (i.e., $z = 0$), there is $F(x) = F(\tilde{x})$, which means ideally the MI-PL operation does not influence the predictions on the clean inputs. When the input is adversarial (i.e., $z = 1$), MI-PL can be applied as a general-purpose defense or a detection-purpose defense, as we separately introduce below:

**General-purpose defense:** If MI-PL can improve the general-purpose robustness, it should satisfy RIC in Eq. (10). By simple derivation and the results of Table 1, this means that

$$\mathbb{E}_{x_s \sim p_s(x|\hat{y})} \left[ G_k(\delta; x_0) - G_k(\lambda\delta; \tilde{x}_0) \right] \begin{cases} > 1 - \lambda, & \text{if } k = \hat{y}, \\ < \lambda - 1, & \text{if } k = y. \end{cases} \tag{12}$$

Since an adversarial perturbation usually suppress the predicted confidence on the true label and promote it on the target label (Goodfellow et al., 2015), there should be $G_{\hat{y}}(\delta; \tilde{x}_0) > 0$ and $G_y(\delta; \tilde{x}_0) < 0$. Note that the left part of Eq. (12) can be decomposed into

$$\underbrace{\mathbb{E}_{x_s \sim p_s(x|\hat{y})} \left[ G_k(\delta; x_0) - G_k(\delta; \tilde{x}_0) \right]}_{\textbf{input transfer}} + \underbrace{\mathbb{E}_{x_s \sim p_s(x|\hat{y})} \left[ G_k(\delta; \tilde{x}_0) - G_k(\lambda\delta; \tilde{x}_0) \right]}_{\textbf{perturbation shrinkage}}. \tag{13}$$

Here Eq. (13) indicates the two basic mechanisms of the MI operations defending adversarial attacks, as shown in Fig. 1(c). The first mechanism is *input transfer*, i.e., the clean input that the adversarial perturbation acts on transfers from the deterministic $x_0$ to stochastic $\tilde{x}_0$. Compared to the Gaussian noise or different image processing methods which introduce spatially or semantically local randomness, the stochastic $\tilde{x}_0$ induces spatially global and semantically diverse randomness. This will make it harder to perform an adaptive attack in the white-box setting (Athalye et al., 2018).

The second mechanism is *perturbation shrinkage*, where the original perturbation $\delta$ shrinks by a factor $\lambda$. This equivalently shrinks the perturbation threshold since $\|\lambda\delta\|_p = \lambda\|\delta\|_p \leq \lambda\epsilon$, which means that MI generally imposes a tighter upper bound on the potential attack ability for a crafted perturbation. Besides, empirical results in previous work also show that a smaller perturbation threshold largely weakens the effect of attacks (Kurakin et al., 2018). Therefore, if an adversarial attack defended by these two mechanisms leads to a prediction degradation as in Eq. (12), then applying MI-PL would improve the robustness against this adversarial attack. Similar properties also hold for MI-OL as described in Sec. 3.2.2. In Fig. 2, we empirically demonstrate that most of the existing adversarial attacks, e.g., the PGD attack (Madry et al., 2018) satisfies these properties.

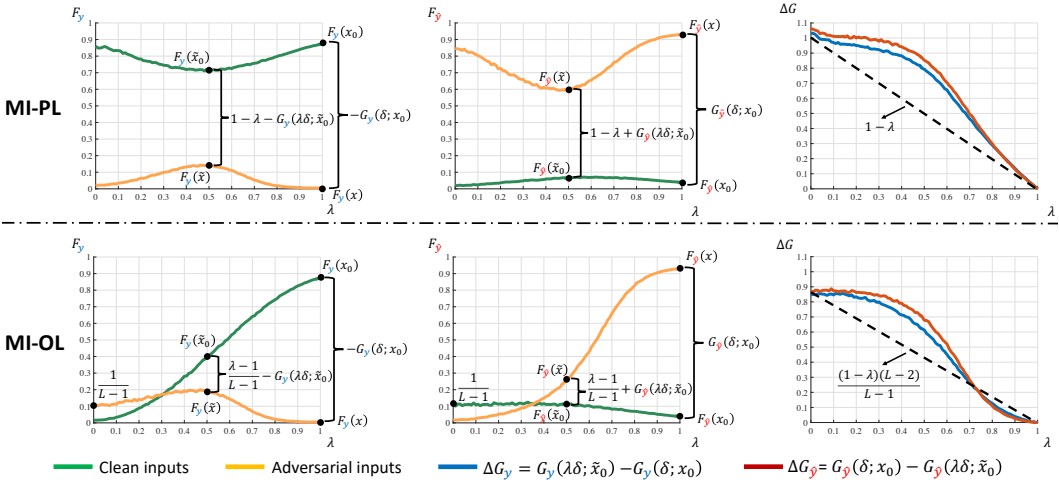

Figure 2: The results are averaged on 100 randomly test clean samples of **CIFAR-10**. The adversarial attack is untargeted PGD-10. Note that the $\Delta G_y$ calculated here is the minus value of it in Eq. (12) and Eq. (15).

**Detection-purpose defense:** According to Eq. (11), the formula of DG for MI-PL is

$$\mathbb{DG}_{\text{MI-PL}} = \mathbb{E}_{x_s \sim p_s(x|\hat{y})}[G_{\hat{y}}(\delta; x_0) - G_{\hat{y}}(\lambda\delta; \tilde{x}_0)] - (1 - \lambda). \tag{14}$$

By comparing Eq. (12) and Eq. (14), we can find that they are consistent with each other, which means that for a given adversarial attack, if MI-PL can better defend it in general-purpose, then ideally MI-PL can also better detect the crafted adversarial examples.

### 3.2.2 MIXUP INFERENCE WITH OTHER LABELS

As to MI-OL, when the input is clean ($z = 0$), there would be a degeneration on the optimal clean prediction as $F_y(\tilde{x}) = F_{\hat{y}}(\tilde{x}) = \lambda$, since the sampled $x_s$ does not come from the true label $y$. As compensation, MI-OL can better improve robustness compared to MI-PL when the input is adversarial ($z = 1$), since the sampled $x_s$ also does not come from the adversarial label $\hat{y}$ in this case.

**General-purpose defense:** Note that in the MI-OL formulas of Table 1, there is a term of $\mathbf{1}_{y=y_s}$. Since we uniformly select $y_s$ from the set $[L] \setminus \{\hat{y}\}$, there is $\mathbb{E}(\mathbf{1}_{y=y_s}) = \frac{1}{L-1}$. According to the RIC, MI-OL can improve robustness against the adversarial attacks if there satisfies

$$\mathbb{E}_{y_s \sim \mathcal{U}_{\hat{y}}(y)} \mathbb{E}_{x_s \sim p_s(x|y_s)} [G_k(\delta; x_0) - G_k(\lambda\delta; \tilde{x}_0)] \begin{cases} > 0, & \text{if } k = \hat{y}, \\ < \frac{(\lambda-1)(L-2)}{L-1}, & \text{if } k = y. \end{cases} \tag{15}$$

Note that the conditions in Eq. (15) is strictly looser than Eq. (12), which means MI-OL can defend broader range of attacks than MI-PL, as verified in Fig. 2.

**Detection-purpose defense:** According to Eq. (11) and Table 1, the DG for MI-OL is

$$\mathbb{DG}_{\text{MI-OL}} = \mathbb{E}_{y_s \sim \mathcal{U}_{\hat{y}}(y)} \mathbb{E}_{x_s \sim p_s(x|y_s)}[G_{\hat{y}}(\delta; x_0) - G_{\hat{y}}(\lambda\delta; \tilde{x}_0)] - (1 - \lambda). \tag{16}$$

It is interesting to note that $\mathbb{DG}_{\text{MI-PL}} = \mathbb{DG}_{\text{MI-OL}}$, thus the two variants of MI have the same theoretical performance in the detection-purpose defenses. However, in practice we find that MI-PL performs better than MI-OL in detection, since empirically mixup-trained models cannot induce ideal global linearity (*cf.* Fig. 2 in Zhang et al. (2018)). Besides, according to Eq. (6), to statistically make sure that the clean inputs will be correctly classified after MI-OL, there should be $\forall k \in [L] \setminus \{y\}$,

$$\mathbb{E}_{y_s \sim \mathcal{U}_{\hat{y}}(y)} \mathbb{E}_{x_s \sim p_s(x|y_s)}[F_y - F_k] > 0 \implies \lambda > L^{-1}. \tag{17}$$

## 4 EXPERIMENTS

In this section, we provide the experimental results on CIFAR-10 and CIFAR-100 (Krizhevsky & Hinton, 2009) to demonstrate the effectiveness of our MI methods on defending adversarial attacks. Our codes are available at https://github.com/P2333/Mixup-Inference.

Table 2: Classification accuracy (%) on the *oblivious* adversarial examples crafted on 1,000 randomly sampled test points of **CIFAR-10**. Perturbation $\epsilon = 8/255$ with step size $2/255$. The subscripts indicate the number of iteration steps when performing attacks. The notation $\leq 1$ represents accuracy less than 1%. The parameter settings for each method can be found in Table 4.

| Methods | Cle. | Untargeted Mode | | | Targeted Mode | | |
|---|---|---|---|---|---|---|---|
| | | $PGD_{10}$ | $PGD_{50}$ | $PGD_{200}$ | $PGD_{10}$ | $PGD_{50}$ | $PGD_{200}$ |
| Mixup | 93.8 | 3.6 | 3.2 | 3.1 | $\leq 1$ | $\leq 1$ | $\leq 1$ |
| Mixup + Gaussian noise | 84.4 | 13.5 | 9.6 | 8.8 | 37.7 | 28.6 | 27.9 |
| Mixup + Random rotation | 82.0 | 21.8 | 18.7 | 18.2 | 38.9 | 32.5 | 26.5 |
| Mixup + Xie et al. (2018) | 82.1 | 23.0 | 19.6 | 19.1 | 38.4 | 31.1 | 25.2 |
| Mixup + Guo et al. (2018) | 83.3 | 31.2 | 28.8 | 28.3 | **57.8** | 49.1 | 48.9 |
| ERM + **MI-OL** (*ablation study*) | 81.6 | 7.4 | 6.4 | 6.1 | 33.0 | 26.7 | 23.2 |
| Mixup + **MI-OL** | 83.9 | 26.1 | 18.8 | 18.3 | 55.6 | **51.2** | **50.8** |
| Mixup + **MI-Combined** | 82.9 | **33.7** | **31.0** | **30.7** | 56.1 | 49.7 | 49.4 |
| Interpolated AT | 89.7 | 46.7 | 43.5 | 42.5 | 65.6 | 62.5 | 61.9 |
| Interpolated AT + Gaussian noise | 84.7 | 55.6 | 53.7 | 53.5 | 70.1 | 69.1 | 69.0 |
| Interpolated AT + Random rotation | 83.4 | 57.8 | 56.7 | 55.9 | 69.8 | 68.2 | 67.4 |
| Interpolated AT + Xie et al. (2018) | 82.1 | 59.7 | 58.4 | 57.9 | 71.1 | 69.7 | 69.3 |
| Interpolated AT + Guo et al. (2018) | 83.9 | 60.9 | 60.7 | 60.3 | 73.2 | 72.1 | 71.6 |
| AT + **MI-OL** (*ablation study*) | 81.2 | 56.2 | 55.8 | 55.1 | 67.7 | 67.2 | 66.4 |
| Interpolated AT + **MI-OL** | 84.2 | **64.5** | **63.8** | **63.3** | **75.3** | **74.7** | **74.5** |

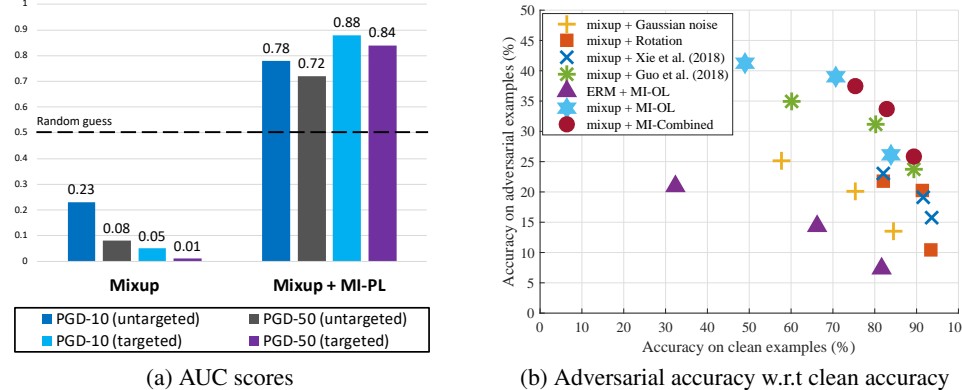

(a) AUC scores · (b) Adversarial accuracy w.r.t clean accuracy

Figure 3: Results on **CIFAR-10**. (a) AUC scores on 1,000 randomly selected test clean samples and 1,000 adversarial counterparts crafted on these clean samples. (b) The adversarial accuracy w.r.t clean accuracy on 1,000 randomly selected test samples. The adversarial attack is untargeted PGD-10, with $\epsilon = 8/255$ and step size $2/255$. Each point for a certain method corresponds to a set of hyperparameters.

## 4.1 SETUP

In training, we use ResNet-50 (He et al., 2016) and apply the momentum SGD optimizer (Qian, 1999) on both CIFAR-10 and CIFAR-100. We run the training for 200 epochs with the batch size of 64. The initial learning rate is 0.01 for ERM, mixup and AT; 0.1 for interpolated AT (Lamb et al., 2019). The learning rate decays with a factor of 0.1 at 100 and 150 epochs. The attack method for AT and interpolated AT is untargeted PGD-10 with $\epsilon = 8/255$ and step size $2/255$ (Madry et al., 2018), and the ratio of the clean examples and the adversarial ones in each mini-batch is $1 : 1$ (Lamb et al., 2019). The hyperparameter $\alpha$ for mixup and interpolated AT is 1.0 (Zhang et al., 2018). All defenses with randomness are executed 30 times to obtain the averaged predictions (Xie et al., 2018).

## 4.2 EMPIRICAL VERIFICATION OF THEORETICAL ANALYSES

To verify and illustrate our theoretical analyses in Sec. 3, we provide the empirical relationship between the output predictions of MI and the hyperparameter $\lambda$ in Fig. 2. The notations and formulas annotated in Fig. 2 correspond to those introduced in Sec. 3. We can see that the results follow our theoretical conclusions under the assumption of ideal global linearity. Besides, both MI-PL and MI-OL empirically satisfy RIC in this case, which indicates that they can improve robustness under the untargeted PGD-10 attack on CIFAR-10, as quantitatively demonstrated in the following sections.

Table 3: Classification accuracy (%) on the *oblivious* adversarial examples crafted on 1,000 randomly sampled test points of **CIFAR-100**. Perturbation $\epsilon = 8/255$ with step size $2/255$. The subscripts indicate the number of iteration steps when performing attacks. The notation $\leq 1$ represents accuracy less than 1%. The parameter settings for each method can be found in Table 5.

| Methods | Cle. | Untargeted Mode | | | Targeted Mode | | |
|---|---|---|---|---|---|---|---|
| | | $PGD_{10}$ | $PGD_{50}$ | $PGD_{200}$ | $PGD_{10}$ | $PGD_{50}$ | $PGD_{200}$ |
| Mixup | 74.2 | 5.5 | 5.3 | 5.2 | $\leq 1$ | $\leq 1$ | $\leq 1$ |
| Mixup + Gaussian noise | 65.0 | 5.5 | 5.3 | 5.3 | 10.0 | 4.3 | 4.1 |
| Mixup + Random rotation | 66.2 | 7.8 | 6.7 | 6.3 | 21.4 | 15.5 | 15.2 |
| Mixup + Xie et al. (2018) | 66.3 | 9.6 | 7.6 | 7.4 | 30.2 | 22.5 | 22.3 |
| Mixup + Guo et al. (2018) | 66.1 | 13.1 | 10.8 | 10.5 | 33.3 | 26.3 | 26.1 |
| Mixup + **MI-OL** | 68.8 | 12.6 | 9.4 | 9.1 | **37.0** | **29.0** | **28.7** |
| Mixup + **MI-Combined** | 67.0 | **14.8** | **11.7** | **11.3** | 31.4 | 26.9 | 26.7 |
| Interpolated AT | 64.7 | 26.6 | 24.1 | 24.0 | 52.0 | 50.1 | 49.8 |
| Interpolated AT + Gaussian noise | 60.4 | 32.6 | 31.6 | 31.4 | 50.1 | 50.0 | 49.6 |
| Interpolated AT + Random rotation | 62.6 | 34.5 | 32.4 | 32.1 | 51.0 | 49.9 | 49.7 |
| Interpolated AT + Xie et al. (2018) | 62.1 | 42.2 | 41.5 | 41.3 | 57.1 | 56.3 | 55.8 |
| Interpolated AT + Guo et al. (2018) | 61.5 | 36.2 | 33.7 | 33.3 | 53.8 | 52.4 | 52.2 |
| Interpolated AT + **MI-OL** | 62.0 | **43.8** | **42.8** | **42.5** | **58.1** | **56.7** | **56.5** |

## 4.3 PERFORMANCE UNDER OBLIVIOUS ATTACKS

In this subsection, we evaluate the performance of our method under the oblivious-box attacks (Carlini & Wagner, 2017). The oblivious threat model assumes that the adversary is not aware of the existence of the defense mechanism, e.g., MI, and generate adversarial examples based on the unsecured classification model. We separately apply the model trained by mixup and interpolated AT as the classification model. The AUC scores for the detection-purpose defense are given in Fig. 3(a). The results show that applying MI-PL in inference can better detect adversarial attacks, while directly detecting by the returned confidence without MI-PL performs even worse than a random guess.

We also compare MI with previous general-purpose defenses applied in the inference phase, e.g., adding Gaussian noise or random rotation (Tabacof & Valle, 2016); performing random padding or resizing after random cropping (Guo et al., 2018; Xie et al., 2018). The performance of our method and baselines on CIFAR-10 and CIFAR-100 are reported in Table 2 and Table 3, respectively. Since for each defense method, there is a trade-off between the accuracy on clean samples and adversarial samples depending on the hyperparameters, e.g., the standard deviation for Gaussian noise, we carefully select the hyperparameters to ensure both our method and baselines *keep a similar performance on clean data* for fair comparisons. The hyperparameters used in our method and baselines are reported in Table 4 and Table 5. In Fig. 3(b), we further explore this trade-off by grid searching the hyperparameter space for each defense to demonstrate the superiority of our method.

As shown in these results, our MI method can significantly improve the robustness for the trained models with induced global linearity, and is compatible with training-phase defenses like the interpolated AT method. As a practical strategy, we also evaluate a variant of MI, called **MI-Combined**, which applies MI-OL if the input is detected as adversarial by MI-PL with a default detection threshold; otherwise returns the prediction on the original input. We also perform ablation studies of ERM / AT + MI-OL in Table 2, where no global linearity is induced. The results verify that our MI methods indeed exploit the global linearity of the mixup-trained models, rather than simply introduce randomness.

## 4.4 PERFORMANCE UNDER WHITE-BOX ADAPTIVE ATTACKS

Following Athalye et al. (2018), we test our method under the white-box adaptive attacks (detailed in Appendix B.2). Since we mainly adopt the PGD attack framework, which synthesizes adversarial examples iteratively, the adversarial noise will be clipped to make the input image stay within the valid range. It results in the fact that with mixup on different training examples, the adversarial perturbation will be clipped differently. To address this issue, we average the generated perturbations over the adaptive samples as the final perturbation. The results of the adversarial accuracy w.r.t the number of adaptive samples are shown in Fig. 4. We can see that even under a strong adaptive attack, equipped with MI can still improve the robustness for the classification models.

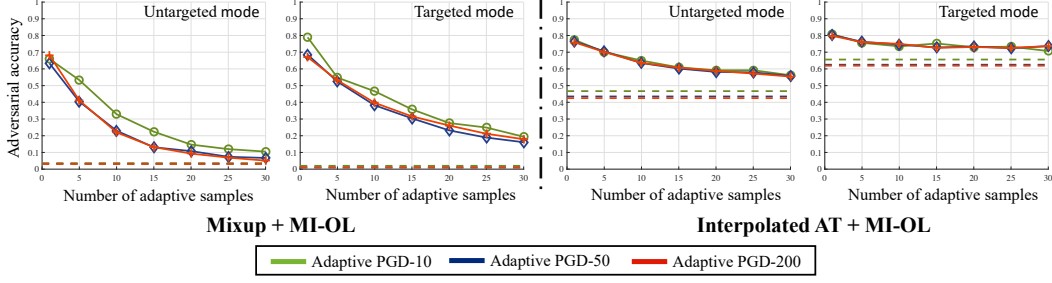

Figure 4: Classification accuracy under the adaptive PGD attacks on **CIFAR-10**. The number of adaptive samples refers to the execution times of sampling $x_s$ in each iteration step of adaptive PGD. The dash lines are the accuracy of trained models without MI-OL under PGD attacks.

## 5 CONCLUSION

In this paper, we propose the MI method, which is specialized for the trained models with globally linear behaviors induced by, e.g., mixup or interpolated AT. As analyzed in Sec. 3, MI can exploit this induced global linearity in the inference phase to shrink and transfer the adversarial perturbation, which breaks the locality of adversarial attacks and alleviate their aggressivity. In experiments, we empirically verify that applying MI can return more reliable predictions under different threat models.

## ACKNOWLEDGEMENTS

This work was supported by the National Key Research and Development Program of China (No. 2017YFA0700904), NSFC Projects (Nos. 61620106010, U19B2034, U1811461), Beijing NSF Project (No. L172037), Beijing Academy of Artificial Intelligence (BAAI), Tsinghua-Huawei Joint Research Program, a grant from Tsinghua Institute for Guo Qiang, Tiangong Institute for Intelligent Computing, the JP Morgan Faculty Research Program and the NVIDIA NVAIL Program with GPU/DGX Acceleration.

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

# A   MORE BACKGROUNDS

In this section, we provide more backgrounds which are related to our work in the main text.

## A.1   ADVERSARIAL ATTACKS AND THREAT MODELS

**Adversarial attacks.** Although deep learning methods have achieved substantial success in different domains (Goodfellow et al., 2016), human imperceptible adversarial perturbations can be easily crafted to fool high-performance models, e.g., deep neural networks (DNNs) (Nguyen et al., 2015).

One of the most commonly studied adversarial attack is the projected gradient descent (PGD) method (Madry et al., 2018). Let $r$ be the number of iteration steps, $x_0$ be the original clean example, then PGD iteratively crafts the adversarial example as

$$x_i^* = \text{clip}_{x,\epsilon}(x_{i-1}^* + \epsilon_i \cdot \text{sign}(\nabla_{x_{i-1}^*} \mathcal{L}(x_{i-1}^*, y))), \tag{18}$$

where $\text{clip}_{x,\epsilon}(\cdot)$ is the clipping function. Here $x_0^*$ is a randomly perturbed image in the neighborhood of $x_0$, i.e., $\mathring{U}(x_0, \epsilon)$, and the finally returned adversarial example is $x = x_r^* = x_0 + \delta$, following our notations in the main text.

**Threat models.** Here we introduce different threat models in the adversarial setting. As suggested in Carlini et al. (2019), a threat model includes a set of assumptions about the adversarys goals, capabilities, and knowledge.

Adversary's goals could be simply fooling the classifiers to misclassify, which is referred to as *untargeted mode*. Alternatively, the goals can be more specific to make the model misclassify certain examples from a source class into a target class, which is referred to as *targeted mode*. In our experiments, we evaluate under both modes, as shown in Table 2 and Table 3.

Adversary's capabilities describe the constraints imposed on the attackers. Adversarial examples require the perturbation $\delta$ to be bounded by a small threshold $\epsilon$ under $\ell_p$-norm, i.e., $\|\delta\|_p \le \epsilon$. For example, in the PGD attack, we consider under the $\ell_\infty$-norm.

Adversary's knowledge describes what knowledge the adversary is assumed to have. Typically, there are three settings when evaluating a defense method:

- *Oblivious adversaries* are not aware of the existence of the defense $D$ and generate adversarial examples based on the unsecured classification model $F$ (Carlini & Wagner, 2017).
- *White-box adversaries* know the scheme and parameters of $D$, and can design adaptive methods to attack both the model $F$ and the defense $D$ simultaneously (Athalye et al., 2018).
- *Black-box adversaries* have no access to the parameters of the defense $D$ or the model $F$ with varying degrees of black-box access (Dong et al., 2018).

In our experiments, we mainly test under the oblivious setting (Sec. 4.3) and white-box setting (Sec. 4.4), since previous work has already demonstrated that randomness itself is efficient on defending black-box attacks (Guo et al., 2018; Xie et al., 2018).

## A.2   INTERPOLATED ADVERSARIAL TRAINING

To date, the most widely applied framework for adversarial training (AT) methods is the saddle point framework introduced in Madry et al. (2018):

$$\min_\theta \rho(\theta), \text{ where } \rho(\theta) = \mathbb{E}_{(x,y)\sim p}[\max_{\delta \in S} \mathcal{L}(x + \delta, y; \theta)]. \tag{19}$$

Here $\theta$ represents the trainable parameters in the classifier $F$, and $S$ is a set of allowed perturbations. In implementation, the inner maximization problem for each input-label pair $(x, y)$ is approximately solved by, e.g., the PGD method with different random initialization (Madry et al., 2018).

As a variant of the AT method, Lamb et al. (2019) propose the interpolated AT method, which combines AT with mixup. Interpolated AT trains on interpolations of adversarial examples along with interpolations of unperturbed examples (*cf.* Alg. 1 in Lamb et al. (2019)). Previous empirical results demonstrate that interpolated AT can obtain higher accuracy on the clean inputs compared to the AT method without mixup, while keeping the similar performance of robustness.

Table 4: The parameter settings for the methods in Table 2. The number of execution for each random method is 30.

| Methods | Parameter Settings |
|---|---|
| Mixup | - |
| Mixup + Gaussian noise | Noise standard deviation $\sigma = 0.04$ |
| Mixup + Random rotation | Rotation degree range $[-40°, 40°]$ |
| Mixup + Xie et al. (2018) | The random crop size is randomly selected from $[16, 24]$ |
| Mixup + Guo et al. (2018) | The random crop size is randomly selected from $[22, 30]$ |
| ERM + **MI-OL** (*ablation study*) | The $\lambda_{\text{OL}} = 0.6$ |
| Mixup + **MI-OL** | The $\lambda_{\text{OL}} = 0.5$ |
| Mixup + **MI-Combined** | The $\lambda_{\text{OL}} = 0.5$, $\lambda_{\text{OL}} = 0.4$, threshold is 0.2 |
| Interpolated AT | - |
| Interpolated AT + Gaussian noise | Noise standard deviation $\sigma = 0.075$ |
| Interpolated AT + Random rotation | Rotation degree range $[-30°, 30°]$ |
| Interpolated AT + Xie et al. (2018) | The random crop size is randomly selected from $[20, 28]$ |
| Interpolated AT + Guo et al. (2018) | The random crop size is randomly selected from $[20, 28]$ |
| AT + **MI-OL** (*ablation study*) | The $\lambda_{\text{OL}} = 0.8$ |
| Interpolated AT + **MI-OL** | The $\lambda_{\text{OL}} = 0.6$ |

# B  TECHNICAL DETAILS

We provide more technical details about our method and the implementation of the experiments.

## B.1  MORE DISCUSSION ON THE MI METHOD

**Generality.** According to Sec. 3, except for the mixup-trained models, the MI method is generally compatible with any trained model with induced global linearity. These models could be trained by other methods, e.g., manifold mixup (Verma et al., 2019a; Inoue, 2018; Lamb et al., 2019). Besides, to better defend white-box adaptive attacks, the mixup ratio $\lambda$ in MI could also be sampled from certain distribution to put in additional randomness.

**Empirical gap.** As demonstrated in Fig. 2, there is a gap between the empirical results and the theoretical formulas in Table 1. This is because that the mixup mechanism mainly acts as a regularization in training, which means the induced global linearity may not satisfy the expected behaviors. To improve the performance of MI, a stronger regularization can be imposed, e.g., training with mixup for more epochs, or applying matched $\lambda$ both in training and inference.

## B.2  ADAPTIVE ATTACKS FOR MIXUP INFERENCE

Following Athalye et al. (2018), we design the adaptive attacks for our MI method. Specifically, according to Eq. (6), the expected model prediction returned by MI is:

$$F_{\text{MI}}(x) = \mathbb{E}_{p_s}[F(\lambda x + (1 - \lambda)x_s)]. \tag{20}$$

Note that generally the $\lambda$ in MI comes from certain distribution. For simplicity, we fix $\lambda$ as a hyperparameter in our implementation. Therefore, the gradients of the prediction w.r.t. the input $x$ is:

$$\frac{\partial F_{\text{MI}}(x)}{\partial x} = \mathbb{E}_{p_s}\left[\frac{\partial F(\lambda x + (1 - \lambda)x_s)}{\partial x}\right] \tag{21}$$

$$= \mathbb{E}_{p_s}\left[\frac{\partial F(u)}{\partial u}\bigg|_{u=\lambda x+(1-\lambda)x_s} \cdot \frac{\partial \lambda x + (1 - \lambda)x_s}{\partial x}\right] \tag{22}$$

$$= \lambda \mathbb{E}_{p_s}\left[\frac{\partial F(u)}{\partial u}\bigg|_{u=\lambda x+(1-\lambda)x_s}\right]. \tag{23}$$

Table 5: The parameter settings for the methods in Table 3. The number of execution for each random method is 30.

| Methods | Parameter Settings |
|---|---|
| Mixup | - |
| Mixup + Gaussian noise | Noise standard deviation $\sigma = 0.025$ |
| Mixup + Random rotation | Rotation degree range $[-20°, 20°]$ |
| Mixup + Xie et al. (2018) | The random crop size is randomly selected from $[18, 26]$ |
| Mixup + Guo et al. (2018) | The random crop size is randomly selected from $[24, 32]$ |
| Mixup + **MI-OL** | The $\lambda_{\mathrm{OL}} = 0.5$ |
| Mixup + **MI-Combined** | The $\lambda_{\mathrm{OL}} = 0.5$, $\lambda_{\mathrm{OL}} = 0.4$, threshold is $0.2$ |
| Interpolated AT | - |
| Interpolated AT + Gaussian noise | Noise standard deviation $\sigma = 0.06$ |
| Interpolated AT + Random rotation | Rotation degree range $[-20°, 20°]$ |
| Interpolated AT + Xie et al. (2018) | The random crop size is randomly selected from $[22, 30]$ |
| Interpolated AT + Guo et al. (2018) | The random crop size is randomly selected from $[24, 32]$ |
| Interpolated AT + **MI-OL** | The $\lambda_{\mathrm{OL}} = 0.6$ |

Figure 5: Adversarial examples crafted by adaptive attacks with $\epsilon = 16/255$ on CIFAR-10, against the defense of Interpolated AT + MI-OL.

In the implementation of adaptive PGD attacks, we first sample a series of examples $\{x_{s,k}\}_{k=1}^{N_A}$, where $N_A$ is the number of adaptive samples in Fig. 3. Then according to Eq. (18), the sign of gradients used in adaptive PGD can be approximated by

$$\mathrm{sign}\left(\frac{\partial F_{\mathrm{MI}}(x)}{\partial x}\right) \approx \mathrm{sign}\left(\sum_{k=1}^{N_A} \frac{\partial F(u)}{\partial u}\Big|_{u=\lambda x+(1-\lambda)x_{s,k}}\right). \tag{24}$$

### B.3 HYPERPARAMETER SETTINGS

The hyperparameter settings of the experiments shown in Table 2 and Table 3 are provided in Table 4 and Table 5, respectively. Since the original methods in Xie et al. (2018) and Guo et al. (2018) are both designed for the models on ImageNet, we adapt them for CIFAR-10 and CIFAR-100. Most of our experiments are conducted on the NVIDIA DGX-1 server with eight Tesla P100 GPUs.

