# OpenReview forum: "Mixup Inference: Better Exploiting Mixup to Defend Adversarial Attacks"
_ICLR.cc/2020/Conference — Accept (Poster)_

### Official Review · AnonReviewer1 · 2019-10-08
**Official Blind Review #1**

**Rating:** 6

**Review:**

Notes:

-Paper claims that adversarial examples stem from locally non-linear behavior.  However, wasn't this the exact opposite of the conclusion of "Explaining and Harnessing Adversarial Examples" (Goodfellow 2015)?  It is cited in this paper but I think the proper conclusion is the opposite of what is written here.

-Linearity however may still be an important component since it simplifies the problem of adversarial robustness.

-Many techniques try to introduce adversarial robustness through transformations during inference time.

-Paper claims that adversarial training induces locally linear behavior - but I'm rather skeptical of this claim.  Think about something like KNN with k=1 and euclidean distance.  This should be L2-robust on the training set, yet very non-linear.

-I understand the contents of Figure 1, but I'm not sure exactly what conclusion I should draw from it.

-The novel procedure presented is "Mixup Inference".  This involves classifying using interpolations of test inputs.

-Two mechanisms are proposed for why this could help.  One is that the magnitude of the perturbation will shrink after doing mixup (although the signal in the original image will also shrink, so I'm not sure if I like this argument).  The second argument is that the adversarial perturbation will have to appear with different random examples which will force the attack to be more "universal" to succeed.  This second argument I find much more compelling.

-The notation $y_s \sim p_s(y_s)$ is rather abusive since the random variable and the same have the same name but this is common in machine learning.

-The technique if I understand correctly (Algorithm 1) amounts to using an average of the prediction at the original point along with an average of the mixes going to all other points in the dataset.

-The paper is a bit slow to explain what distribution lambda will come from - but it effects the algorithm a lot (especially if the distribution is symmetric or asymmetric).

Comments:

-There is another paper (Shimada 2019) that also uses interpolation at test time and should be cited here, although I admit that paper is written in a confusing way so the connection may not be immediately obvious: https://arxiv.org/pdf/1906.08412v1.pdf

-I have a suggestion for the organization of the paper that I think would improve it.  I would suggest to first introduce the method in a clear fashion (after motivating it), along with the equations.  Then, *after that*, clearly and separately introduce the analysis of the "optimal linear model": "a well mixup-trained model F can be denoted as a linear function H on the convex combinations of clean examples".  I think that would make the paper much clearer.

-For example, how can we actually know what G_k() is unless we know the adversarial perturbation (which shouldn't generally be possible during inference)?  I found this discussion to be rather confusing (basically section 3.1) although admittedly it might be my own fault.

-Why does mixup-inference hurt the clean accuracy by 10% (table 2)?  This seems like quite a lot to me.  Still the degree of robustness does seem impressive.  And I also believe that the obtained robustness of this technique along with AT is state of the art.

-It might be nice to see examples of attacks on the resulting model, especially the one with the best robustness.  It's possible that Linf-bounded attacks against *this model* will be perceptible, which would support lowering the epsilon-attack-budget (which is actually a good thing for the research field as it's evidence that maybe this epsilon shouldn't be considered practically imperceptible).

Review:

This paper was interesting, because it has nice experimental results and seems like a good idea, but I feel like the paper needs to be improved.  The biggest issue is that the paper repeatedly claims that adversarial robustness can be improved by making networks more linear, yet I believe that this is the opposite of what prior work has found.  I also found the exposition of the idea to be confusing as it simultaneously introduces the idea and an analysis of the technique - I would much prefer if the technique were introduced first and the analysis under some optimal assumptions moved to a different section.

**Experience Assessment:**

I have published in this field for several years.

**Review Assessment: Checking Correctness Of Derivations And Theory:**

I assessed the sensibility of the derivations and theory.

**Review Assessment: Checking Correctness Of Experiments:**

I assessed the sensibility of the experiments.

**Review Assessment: Thoroughness In Paper Reading:**

I read the paper at least twice and used my best judgement in assessing the paper.

---

> ### Author Response · Authors · 2019-11-06
> **Thank you for the valuable review**
>
> Thank you for the valuable review.
>
> We upload the revision according to your suggestions, which includes:
> 1. To be more rigorous, we reclaim that the adversarial examples mainly root from locally unreasonable or unstable behavior, and the adversarial training method induces locally stable behavior.
>
> 2. We check the paper of Schimada et al. (2019) and find that it is indeed a quite related paper. We cite it and mention the difference between our work in Introduction.
>
> 3. We modify the organization as you suggested in Section 3, i.e., we first introduce the method and then provide the theoretical analysis. We also explain the distribution of lambda ahead.
>
> 4. We modify the notations like $y_{s}\sim p_{s}(y_{s})$ to $y_{s}\sim p_{s}(y)$, and $x_{s}\sim p_{s}(x_{s}|y_{s})$ to $x_{s}\sim p_{s}(x|y_{s})$.
>
> 5. We provide adversarial examples crafted by adaptive attacks against the IAT + MI (best robustness) in Figure 5 (in Appendix part).
>
>
>
> Below we further clarify the remain questions.
>
> Question 1. About the linearity:
> As to the relationship between linearity and robustness, actually there is not a final conclusion. Goodfellow et al. (2015) claimed that the vulnerability of neural networks comes from their linear nature mainly because they found that FGSM can successfully attack some networks, and FGSM is based on the linear assumption of classifiers. However, there are many cases where FGSM fails but iterative attacks like PGD can still evade the models, which means adversarial examples do not necessarily stem from the linear nature. Besides, recent work [*1] finds that linearity can improve robustness, which contradicts the conclusion in Goodfellow et al. (2015). So as you suggested, in the revision, we do not claim the relationship between model linearity and the existence of adversarial examples.
>
> [*1] Qin et al. Adversarial Robustness through Local Linearization. NeurIPS 2019
>
>
> Question 2. The shrinkage mechanism in MI:
> It is true that the signal in the original image will also shrink after performing MI. Intuitively, we intend to improve the ‘signal-to-noise ratio’ fed into the classifier, so what we expect is that the effect of adversarial noise shrinks faster than it caused by the original signal. This is actually the property represented in Eq.(12) and Eq.(15), where the left parts of the inequalities indicate the change of the adversarial effect, and the right part indicates the change of the original clean effect.
>
>
> Question 3. About the function $G_{k}$:
> We cannot have a closed-form representation on $G_{k}$, since the neural network is a black-box model. So what we can claim in the theoretical analyses is that MI can improve robustness on the attacks satisfying Eq.(12) and Eq.(15), i.e., the decay of the adversarial effect is enough to compensate the decay of the original clean effect. Then in Figure 2, we empirically show that strong attacks like PGD satisfy this property and such that MI can better defend them.
>
>
>
> Answers on other detailed notes and comments:
> (1)  In Algorithm 1, the average is performed on the $N$ randomly sampled points in the dataset with label $y_{s}$. In the experiments, we follow the setting in previous work (Xie et al. ICLR 2018) to set $N=30$ for fair comparisions.
>
> (2)  In Table 2, we choose the hyperparameter setting with comparable clean accuracy for each method to perform a fair comparison. As shown in Figure 3(b), there is a trade-off between clean accuracy and adversarial accuracy depending on the hyperparameters in each method, and MI can lead to better trade-off.

---

> > ### Comment · AnonReviewer1 · 2019-11-14
> > **Response to Rebuttal**
> >
> > "As to the relationship between linearity and robustness, actually there is not a final conclusion. Goodfellow et al. (2015) claimed that the vulnerability of neural networks comes from their linear nature mainly because they found that FGSM can successfully attack some networks, and FGSM is based on the linear assumption of classifiers. However, there are many cases where FGSM fails but iterative attacks like PGD can still evade the models, which means adversarial examples do not necessarily stem from the linear nature. Besides, recent work [*1] finds that linearity can improve robustness, which contradicts the conclusion in Goodfellow et al. (2015). So as you suggested, in the revision, we do not claim the relationship between model linearity and the existence of adversarial examples."
> >
> > So I think that the older informal claim is that non-linearity is the *cause* of adversarial examples.  Goodfellow 2015 argued against this by showing that linear models also can have adversarial examples (but of course they can also be robust).  So I don't think that "only linear models can have adversarial examples" is the right conclusion to draw from Goodfellow 2015.
> >
> > I'm still okay on this point as long as the paper doesn't claim anything misleading in the introduction, as I don't see it as essential to the paper.  As such I'm happy with the introduction in its current form.
> >
> > Point 2 makes sense to me and I really like the re-organization done for point 3.

---

> > > ### Author Response · Authors · 2019-11-14
> > > **Thank you!**
> > >
> > > Thank you again for your kind suggestions, which really help a lot to improve the original version of the paper. We deeply appreciate it!

---

### Official Review · AnonReviewer3 · 2019-10-23
**Official Blind Review #3**

**Rating:** 6

**Review:**

This paper proposes a novel use of mixup, which is originally a data augmentation method incorporating two training samples and their corresponding labels. The authors utilize mixup not for training but for inference (MI; Mixup Inference). Experimental results on Cifar 10, and Cifar 100 show that MI can boost the classification performance in combination with interpolated AT (Adversarial  Training) and mixup.

I lean to accept this paper. The proposed method is simple but effective, moreover well-motivated. The experimental results, including several ablation studies, show a high versatility with existing methods.

My minor concerns are, however, consisting of two points.
- The authors should repeat the experiments several times and show the averages and standard errors to make the significance clear.
- Both Cifar 10 and Cifar 100 are relatively small scale datasets. I would like the authors to investigate larger ones.

**Experience Assessment:**

I have published in this field for several years.

**Review Assessment: Checking Correctness Of Derivations And Theory:**

I assessed the sensibility of the derivations and theory.

**Review Assessment: Checking Correctness Of Experiments:**

I carefully checked the experiments.

**Review Assessment: Thoroughness In Paper Reading:**

I read the paper thoroughly.

---

> ### Author Response · Authors · 2019-11-06
> **Thank you for the supportive review**
>
> Thank you for the supportive review.
>
> Showing the average and standard errors is a good suggestion, we are running the experiments and will add them in the final version.
>
> We also have some initial results on ImageNet, where applying MI can improve the adversarial accuracy of mixup-trained models from 3.6% to 35%, under PGD-10 attacks with $\epsilon=4/255$.

---

> ### Author Response · Authors · 2019-11-11
> **More results on ImageNet**
>
> We train a Resnet-50 with the mixup on ImageNet, and the top-1 accuracy on 50,000 validation examples is 73.7%. In the adversarial setting, we test on 1,000 randomly selected examples in the validation set. The accuracy results (%) on the sampled validation subset are shown below:
>
>         Method     || Clean Acc| $\text{PGD}_{10}^{\textbf{tar}}$ |$\text{PGD}_{10}^{\textbf{un}}$ | $\text{PGD}_{50}^{\textbf{tar}}$|$\text{PGD}_{50}^{\textbf{un}}$ ||
>          Mixup       ||      88.3     |      1.3      |      3.6     |      0.5      |      3.4    ||
> Mixup + MI-OL ||      82.2     |      67.2    |     35.0   |      64.1    |     24.6   ||
>
> Here $\epsilon=4/255$, number of execution in MI is $N=30$, and $\lambda_{\text{OL}}=0.4$.

---

### Official Review · AnonReviewer2 · 2019-10-26
**Official Blind Review #2**

**Rating:** 6

**Review:**

This paper introduces a novel method for an adversarial attack named mixup inference (MI).  Most of the work focuses on embedding mixup mechanism in the training phase, but MI uses the mixup in the inference phase. MI method has two main effects for the adversarial attack: one is perturbation shrinkage, and the other one is input transfer because MI can exploit
the induced global linearity. The experimental results show that MI can return more reliable predictions under different threat models.

This paper should be accepted because the proposed method is super simple but effective for defending from adversarial attacks under different threat conditions. This paper is well-written, including theoretical insights on why the MI method works.

The reviewer has some questions or comments to clarify the paper:
1) In the explanation of the MI method, the authors assume only the cases where the input data is correctly classified if it is clean, or wrongly classified if it is adversarial. In a realistic situation, the classifier sometimes outputs mislabels. Thus is the discussion in Sec.3 valid if the clean input data misclassified?

2) To predict the category of the input, MI methods must perform inference N times. It is not efficient. Are there any ideas to reduce the number of inferences?

3) MI-Combined seems ad-hoc. It would be better to state its justification by theory.

4) The same idea of mixup was proposed at the same conference (ICLR2018). It should be cited.
Tokozume et al., Learning from Between-class Examples for Deep Sound Recognition. ICLR, 2018.

**Experience Assessment:**

I have published in this field for several years.

**Review Assessment: Checking Correctness Of Derivations And Theory:**

I assessed the sensibility of the derivations and theory.

**Review Assessment: Checking Correctness Of Experiments:**

I assessed the sensibility of the experiments.

**Review Assessment: Thoroughness In Paper Reading:**

I read the paper at least twice and used my best judgement in assessing the paper.

---

> ### Author Response · Authors · 2019-11-06
> **Thank you for the supportive review**
>
> Thank you for the supportive review.
>
> Question 1. The case of mislabeling:
> The improvement on decreasing output mislabels is mainly finished in the training phase by mixup or other better training mechanisms. The main effect of MI is to improve adversarial robustness, where the original clean counterpart is correctly classified. It is possible that MI can correctly classify the adversarial examples crafted on mislabeled clean images, but this is not guaranteed by theoretical analyses.
>
>
> Question 2. More efficient inference:
> MI can still have good performance when $N=1$. A higher number of $N$ can stabilize the inference progress, but is not necessary. In our experiment, we choose $N=30$ to follow the setting in previous work (Xie et al. ICLR 2018).
>
>
> Question 3. About MI-Combined:
> The MI-Combined method consists of MI-PL in the detection phase and MI-OL in the classification phase. These two phases are separately justified in Eq.(14) and Eq.(15).
>
>
> Question 4. Related work:
> Thank you for pointing out, we have added the reference of Tokozume et al. (ICLR 2018) in the uploaded revision.

---

### Public Comment · ~Bao_Wang1 · 2019-10-19
**An interesting paper**

Hi, I read this paper which is quite interesting to me. I would like to point out three papers that among the first a few that considered interpolation for adversarial defense.

1. B. Wang, et al. Deep Neural Nets with Interpolating Function as Output Activation, NeurIPS 2018.

2. B. Wang, et al. Adversarial Defense via Data Dependent Activation Function and Total Variation Minimization, arXiv:1809.08516 2018

3. B. Wang, et al. Graph Interpolating Activation Improves Both Natural and Robust Accuracies in Data-Efficient Deep Learning, arXiv:1907.06800 2019.

Thanks for your attention.

---

> ### Comment · AnonReviewer1 · 2019-10-19
> **Inference or Training?**
>
> Were the papers you linked to using mixup during training or inference?  I believe that this paper's focus is on using mixup at inference-time for robustness.
>
> It may still be the case that those papers should be cited.

---

> > ### Public Comment · ~Bao_Wang1 · 2019-10-26
> > **Both inference and training**
> >
> > Thanks! The previous papers used interpolation steps in both training and inference to improve robustness of the deep neural nets. Not only in the semi-supervised learning but also in the supervised learning. I believe the papers are very related, and these papers are among the first few papers that considered interpolation in deep nets.

---

> ### Author Response · Authors · 2019-10-20
> **The linked papers seem not quite related**
>
> Thank you for your recommendation on these interesting work, we read the linked papers to check if they are related to our work. As indicated by Reviewer1, our paper focuses on using mixup at inference time for better robustness.
>
> First of all, we find that none of the three linked papers is based on the mixup method (Zhang et al. ICLR 2018), or mentions the mixup method in their related work.
>
> The three linked papers propose and analyze one similar method, named weighted nonlocal Laplacian (WNLL) layer as the output layer for DNNs. The claimed advantages of the WNLL layer is mainly on data-efficient learning and robustness in the semi-supervised setting, while we consider supervised learning. Besides, the WNLL layer benefits the learning in the training phase, while our mixup-inference (MI) method further improves the robustness of mixup-trained models in the inference phase.
>
> Thanks again for your kind suggestions, but it seems that the linked papers are not quite related to our work. Please feel free to let us know if we misunderstand some parts of your linked papers.

---

### Author Response · Authors · 2019-11-14
**Looking forward to further feedbacks**

Dear Reviewers,

Thank you again for your valuable comments and suggestions, which are really helpful for us. We have uploaded new revisions and posted responses to the proposed concerns and questions.

We totally understand that this is a quite busy period of time, since the reviewers may be preparing the rebuttal for their own submissions or rushing for the deadline of the recent conferences.

So we deeply appreciate it if the reviewers can take some time to return further feedbacks on whether our responses and extra experiment results solve their concerns. If there is any other question, we will try our best to provide satisfactory answers.

Best,
The authors

---

### Decision · Program_Chairs · 2019-12-19

**Decision:**

Accept (Poster)

**Comment:**

This paper proposed a mixup inference (MI) method, for  mixup-trained models, to better defend adversarial attacks.  The idea is novel and is proved to be effective on CIFAR-10 and CIFAR-100.  All reviewers and the AC agree to accept the paper.